# Increased Selectivity of Novozym 435 in the Asymmetric Hydrolysis of a Substrate with High Hydrophobicity Through the Use of Deep Eutectic Solvents and High Substrate Concentrations

**DOI:** 10.3390/molecules24040792

**Published:** 2019-02-22

**Authors:** Yerko Fredes, Lesly Chamorro, Zaida Cabrera

**Affiliations:** School of Biochemical Engineering, Pontificia Universidad Católica de Valparaíso; Avda. Brasil 2085 Valparaíso, Chile; y.a.fredes@gmail.com (Y.F.); leslychamorromolina@gmail.com (L.C.)

**Keywords:** asymmetric synthesis, Novozym 435, deep eutectic solvents, dimethyl-3-phenylglutarate, hydrophobic substrates

## Abstract

The effects of the reaction medium and substrate concentration were studied on the selectivity of Novozym 435 using the asymmetric hydrolysis of dimethyl-3-phenylglutarate as a model reaction. Results show that the use of choline chloride ChCl:urea/phosphate buffer 50% (*v/v*) as a reaction medium increased the selectivity of Novozym 435 by 16% (e.e = 88%) with respect to the one in 100% phosphate buffer (e.e = 76%). Best results were obtained when high substrate concentrations (well above the solubility limit, 27-fold) and ChCl:urea/phosphate buffer 50% (*v/v*) as reaction medium at pH 7 and 30 °C were used. Under such conditions, the *R*-monoester was produced with an enantiomeric purity of 99%. Novozym 435 was more stable in ChCl:urea/phosphate buffer 50% (*v/v*) than in phosphate buffer, retaining a 50% of its initial activity after 27 h of incubation at pH 7 and 40 °C. Results suggest that the use of deep eutectic solvents (ChCl:urea/phosphate buffer) in an heterogeneous reaction system (high substrate concentration) is a viable and promising strategy for the synthesis of chiral drugs from highly hydrophobic substrates.

## 1. Introduction

Currently, the enantiomeric purity of a chiral drug is an essential requirement for its commercialization, because even though both optical isomers are chemically similar, the organism is able to distinguish between both enantiomeric forms and recognize only one of them [1,2,3]. Thus, the physiological responses in the organism will be quite different depending on the isomer consumed, and in some cases, they could have opposite effects. In this context, the pharmaceutical industry urgently requires the development of novel production strategies for the synthesis of pure enantiomers, taking into consideration the viability of the process at the industrial level and its environmental benignity. 

Among the available strategies, the asymmetric synthesis of prochiral compounds by enzyme biocatalysis is very promising, by producing essentially pure enantiomers in a single reaction step [4,5]. However, the substrates generally used in this type of reaction are highly hydrophobic, which considerably affects the productivity of the process due to the low substrate concentrations present in the reaction medium, precluding its application at an industrial level. The use of organic solvents allows increasing the solubility of hydrophobic substrates, but notoriously affects the stability of the enzyme [6,7,8].

Within this scenario, it is of great interest to study new reaction medium and/or production strategies that are gentle with the enzyme, allowing high activity and stability, while increasing the solubility of the substrate, with the consequent increase in productivity. In this context, Illanes et al. [9] studied the effect of high concentrations of substrates on the synthesis of cephalexin with immobilized penicillin acylase. Working in a heterogeneous system with partially-undissolved substrates, a three-fold increase in the specific productivity of the process was obtained with respect to a homogeneous system in which substrates were completely dissolved in the reaction medium. Along with this, the stability of the biocatalyst was significantly improved, maintaining 87% of the initial activity of the enzyme after ten consecutive production batches. In this context, Illanes et al. noted that it is clear that, as a consequence of Michaelian kinetics, productivity should increase with increasing substrate concentrations. The factor of the increase depends, however, on the specific catalyst format used, for example under heterogeneous reaction conditions. Moreover, use of heterogeneous catalyst format may affect other properties of the enzyme performance, as enantiomeric excess [9].

On the other hand, Jacobsen et al. evidenced that increasing substrate concentration may reduce the selectivity of enzyme, as shown by a decrease of enantiomeric excess from 91% to 85% in the hydrolysis of diethyl-3-hydroxyglutarate catalyzed by Novozym 435 [10]. This is one of the few reports in which the effect of substrate concentration on the selectivity of an enzyme was evaluated. Therefore, even though the use of high substrate concentrations in the asymmetric synthesis of chiral compounds can improve clearly the productivity of the process, its effect on the selectivity of enzyme is a point that has not yet been adequately explored.

In the line of improving the catalytic properties of enzymes, the use of neoteric solvents, such as deep eutectic solvents (DESs), have unique characteristics that make them attractive as reaction medium for catalytic processes. DESs, like ionic liquids, have almost zero volatility and high thermal stability, but unlike them, DESs are biodegradable and can be synthesized at very low cost [11]. DESs based on choline chloride (ChCl) with organic salt and glycerol, ethylene glycol or urea as the hydrogen bridge donor, have been reported as having increased the activity, selectivity and stability of different lipases [11,12,13,14,15,16]. This background information supports the use of DESs as an interesting reaction medium for biocatalysis with highly hydrophobic substrates [16].

The substrate dimethyl-3-phenylgutarate (DMFG), used as a model for this study (Figure 1), is highly hydrophobic, having low solubility in both aqueous medium and in organic solvents [17]. The synthesis of chiral derivatives from DMFG is of great interest for the pharmaceutical industry, as both enantiomers are important intermediates in the synthesis of several drugs [18,19,20]. In this reaction, neither the substrate (diester) nor the final product (diacid) are chiral compounds. However, the monoester, an intermediate product, is chiral. If the reaction stops at the monoester and the enantioselectivity value of the process is very high, 100% yield of an enantiomerically pure compound will be produced [21].

Several authors have studied the synthesis of these chiral monoesters from different prochiral substrates. In some cases, high specific productivities were obtained, but at the expense of selectivity [10,22], while in other cases, high selectivity was obtained at the expense of specific productivity [16,17,21,23,24].

The objective of this work was to study the effects of the reaction medium (DESs) and substrate concentration on the activity and selectivity of Novozym 435, with the aim of increasing the specific productivity of reactions where hydrophobic substrates are used. The asymmetric hydrolysis of dimethyl-3-phenylglutarate mediated by Novozym 435 was used as a study model. The proposed strategy can contribute to the development of more efficient and economical process for the synthesis of chiral drugs.

## 2. Results and Discussion

ChCl:urea and ChCl:glycerol were previously selected to perform this investigation, because they are within simples DES most utilized with CALB lipase (immobilized and soluble) [11,15]. However, CHCl-glycerol was subsequently discarded because it showed low substrate solubility (0.87 mM of DMFG) in DES-buffer mixture compared with ChCl:urea (2.02 mM) (results not shown). ChCl:urea has been described as more effective for lipase-catalyzed reaction [25].

### 2.1. Asymmetric Hydrolysis of DMFG Catalyzed by Novozym 435

Figure 2 shows the reaction courses of the asymmetric hydrolysis of DMFG catalyzed by Novozym 435 in two reaction media: 25 mM sodium phosphate (onwards, phosphate buffer) at pH 7 and ChCl:urea/phospate buffer 50% (*v/v*) (onwards, ChCl:urea 50% (*v/v*)) at pH 7. Figure 2a corresponds to the homogenous reaction system (dissolved substrate), while Figure 2b corresponds to the heterogeneous reaction system (partially undissolved substrate). In both cases (homogenous and heterogeneous systems), initial specific reaction rate and conversion did not vary significantly from one reaction medium to the other and conversions close to 100% were obtained. The same effect was evidenced by Wu et al., where the enzyme remained the same as active as in the DES-free solution [26].

Table 1 summarizes the main results obtained in the asymmetric hydrolysis of DMFG in terms of specific initial reaction rate, enantiomeric excess (e.e) and specific productivity in different systems and reaction mediums. In both reaction mediums and according to Figure 2, there are no significant differences in the specific initial reaction rate when the same concentration of DMFG was utilized. In addition, in both reaction mediums the reaction rate increased by a factor of 3 when a heterogeneous reaction system was used. The highest value of specific initial reaction rate (6.2 μmol of product min^−1^ g catalyst^−1^) was obtained in the medium ChCl:urea 50% (*v/v*) at 18 mM of DMFG at pH 7 and 30 °C.

The presence of 50% (*v/v*) of DES in the reaction medium, under both homogeneous and heterogeneous systems, considerably improved the results of selectivity obtained with respect to the medium without DES. Novozym 435 increased its selectivity by 16% obtaining *R*-MFG with an e.e of 88% when going from a homogeneous (0.45 mM substrate concentration) to heterogeneous system (18 mM substrate concentration) in a phosphate buffer medium. The selectivity value obtained in the phosphate buffer (e.e = 76%) agrees with the results obtained previously by Cabrera et al. [27]. 

The same trend was evidenced in the reaction medium composed of ChCl:Urea 50% (*v/v*) at pH 7, where the behavior of Novozym 435 was improved in the heterogeneous system (18 mM substrate concentration), obtaining a close to 10% increase in e.e with respect to the homogeneous system (0.45 mM substrate concentration), reaching a value of 94%. 

Improvements in the selectivity of different enzymes and whole cells by the use of DESs as medium of reaction have been reported [11]. In particular, Capriati et al. reported that baker’s yeast exhibits an important change in the rate of reaction and enantioselectivity in the reduction of arylpropanones when changing the solvent from pure water to DES-water mixtures [28]. On the other hand, the effect of the substrate concentration on the selectivity of the enzymes and cells has been scarcely reported in the literature. Jacobsen and collaborators showed that an increase of the concentration decreases the selectivity of the enzyme in the asymmetric hydrolysis of diethyl 3-hydroxyglutarate. However, in the present work, an increase of substrate concentration exerts a positive effect on the selectivity of Novozym 435. We believe that this may be related to the type of molecule and its hydrophobicity. DMFG has a phenyl group that confers a highly hydrophobic character to the molecule; therefore, its interaction with the active site of the enzyme can be very different from that of diethyl 3-hydroxyglutarate. In this context, both the modification of the type of substrate and that of the solvent in the reaction medium, have been simple methods to obtain changes in the selectivity of an enzyme [29].

In terms of specific productivity, the best results were obtained for the heterogeneous system in the ChCl:urea 50% (*v/v*) reaction medium, where the productivity was three times higher than that obtained in the homogeneous system in phosphate buffer medium.

In all cases the *R* isomer of the monoester was obtained. This is characteristic for CALB lipase [27], other enzymes such as BTL2 lipase or Lecitase Ultra phospholipase obtain as product the monoester with S configuration (*S*-methyl-3-phenyl glutarate) [30,31].

In this work the presence of diacid was negligible (less than 2%). We believe that the hydrophobicity of the biocatalyst (Novozym 435) facilitated the partition of the monoester outside the active site of enzyme. This, based on the work of Fernandez-Lorente et al. where the hydrolysis of a dicarboxylic diester promoted generation of a free carboxylic acid, which at pH 7 will have a negative charge. This charged compound will be partitioned from the active site of an enzyme, if the latter presents a highly hydrophobic environment. This appears to virtually stop the reaction at the level of monoester [32].

### 2.2. Effect of High Substrate Concentration on the Selectivity of Novozym 435

In order to improve the previous results further, the effect of substrate concentration was studied at values higher than 18 mM. Table 2 shows that an increase in the substrate concentration in the reaction medium generated a positive effect on the selectivity of Novozym 435, obtaining *R*-MFG with a very high optical purity (e.e = 99%) at 54 mM of DMFG.

Specific productivity was improved by 30% compared to the experience with 18 mM of substrate. This moderate increase could be explained by the oversaturation zone in which the enzyme is found. A value of 0.68 g of product/g biocatalyst/day at 54 mM of DMFG was the maximum specific productivity reached, being five times higher than that obtained in the homogeneous system in phosphate buffer medium.

Results show that e.e can be increased by increasing substrate concentration, which is an asset that can still be properly exploited in asymmetric hydrolysis reactions.

### 2.3. Solubility of R-MFG in ChCl:Urea 50% (v/v)

In order to evaluate the recovery of the chiral product, the solubility of *R*-MFG in the ChCl:urea 50% (*v/v*) reaction medium was determined. Figure 3 shows the production kinetics of *R,S*-MFG catalyzed by Novozym 435. When the reaction was carried out at a substrate concentration of 36 and 54 mM, the maximum concentration of *R*-MFG in the soluble phase was 36 and 37 mM respectively. Such a concentration is quite high when compared to the maximum solubility of DMFG at the same reaction conditions (2.02 mM). This effect may be related to the fact that the monoester has an acid group that is ionized at pH 7. This favors its solubility, being 18.5 times higher than DMFG. 

These results show that it is possible to design production strategies that maximize product recovery, making the process more profitable.

### 2.4. Stability of Novozym 435 in the Presence of Phosphate Buffer and ChCl:Urea 50% (v/v)

The stability of Novozym 435 in non-reactive conditions at 40 °C was evaluated. Figure 4 shows the courses of the inactivation of Novozym 435 in two reaction media: phosphate buffer at pH 7 and ChCl:urea 50% (*v/v*) at pH 7. The stability was performed at 40 °C to force the system and see the stability relationship between buffer and DES-buffer. Figure 4 show that Novozym 435 was more stable in ChCl:urea 50% (*v/v*) than in phosphate buffer, retaining a 50% of its activity at 27 h of incubation at 40 °C. These results agree with those obtained by Homman et al. where under the operating conditions (phosphate buffer at pH 8 and 40 °C) immobilized CALB preparation (Chirazyme L-2) lost 30% of its original activity within the first 18 h [14]. It is important to consider that these values turn out to be much higher under reactive conditions, due to the effect of substrate protection.

## 3. Materials and Methods 

### 3.1. General 

Novozym 435 was kindly donated by Blumos S.A (Santiago, Chile). *p*-nitrophenyl butyrate (*p*NPB) was obtained from Sigma–Aldrich (St. Louis, MO, USA). Dimethyl-3-phenylglutarate was synthesized according to the protocol described by Cabrera et al. (2009) [23]. Choline chloride (ChCl), combined with urea (U), ChCl:urea, 1:2 mol:mol, was kindly donated by professor Carlos Carlessi of the School of Chemical Engineering, Pontificia Universidad Católica de Valparaíso [16]. Other reagents were of analytical or HPLC grade.

### 3.2. Lipase Activity Determination

Activity assay was performed by measuring the increase in absorbance at 348 nm produced by the release of *p*-nitrophenol in the hydrolysis of 0.4 mM *p*NPB in 25 mM sodium phosphate buffer at pH and 30 °C, using a thermostated spectrophotomer with magnetic stirring [23]. To start the reaction, 20 mg of immobilized enzyme were added to 60 mL of substrate solution. An international unit (IU) of *p*NPB activity is defined as the amount of enzyme that hydrolyzes 1 μmol of *p*NPB per minute at the conditions described above.

### 3.3. Asymmetric Hydrolysis of DMFG Catalyzed by Novozym 435

The behavior of Novozym 435 in the hydrolysis of DMFG in two different reaction media: 25 mM sodium phosphate (phosphate buffer) and ChCl:urea/phosphate buffer 50% (*v/v*) (ChCl:urea 50% (*v/v*) and at different substrate concentrations was determined by adding 0.086 g of catalyst in 30 mL of reaction medium at pH 7, 30 °C and 170 rpm. 

Substrate concentrations used in the homogenous reaction system were 0.45 mM for sodium phosphate buffer (maximum solubility of DMFG in the system) and 0.45 mM for ChCl:urea/phosphate buffer 50% (*v/v*), where 2.02 mM is the maximum solubility of DMFG in that system. For the reactions in the heterogeneous system, the DMFG concentration used was 18 mM for sodium phosphate buffer and 18, 36 and 54 mM for ChCl:urea/phosphate buffer 50% (*v/v*). In all cases, triplicates of each assay were done. In case of homogeneous system, the samples, taken at different time intervals, were centrifuged at 4 °C and 10,000 rpm for 10 min (to remove the biocatalyst) and diluted appropriately to determine their substrate concentration by HPLC. In the case of a heterogeneous system, the samples with solids in suspension, also taken at different time intervals, were centrifuged at 4 °C and 10,000 rpm for 15 min. The precipitated (substrate more biocatalyst) was re-suspended in 100% acetonitrile and again centrifuged to the same conditions to remove the biocatalyst. To determine the total concentration of DMFG in the reaction medium, the liquid phase (supernatant) and the solid phase (resuspended precipitate) were appropriately diluted and its substrate concentration by HPLC was determined. The total concentration of DMFG is determined by mass of product in the soluble phase plus mass of product in the solid phase.

The degree of hydrolysis (DMFG decrease) was followed by reverse-phase HPLC (Spectra Physic SP 100 coupled with an UV detector Spectra Physic SP 8450) on a Waters Symmetry column C18 5 μm (4.6 × 150 mm). Elution was performed with a mobile phase composed by acetonitrile (35% *v/v*) and 10 mM ammonium phosphate pH 3 (65% *v/v*), at a flow rate of 1 mL/min. Elution was monitored by recording the absorbance at 225 nm [21]. Concentration of substrate was calculated from calibration curves using stock solutions. Conversion was calculated by using the following equation:Conversion (%) = ((DMFG)_t_/(DMFG)_o_) × 100,
where (DMFG)_t_ is the total substrate concentration (mM) at t hours and (DMFG)_o_ is the total substrate concentration at zero hours (initial concentrations of DMFG (mM)).

Initial reaction rates were calculated from the initial slope of the product versus time curve.

Specific productivity (g product/d g biocatalyst) was determined as the amount of *R*-MFG produced per unit time and unit mass of biocatalyst at maximum conversion.

### 3.4. Determination of Enantiomeric Excess

The enantiomeric excess (e.e) of the *R*-methyl-3-phenyl glutarate (*R*-MFG) formed was analyzed by Chiral Reverse Phase HPLC [21]. The column was a Chiracel OD-R (Chiral Technologies, Inc. West Chester, NY, USA). The mobile phase was acetonitrile (25% *v/v*) and 10 mM ammonium phosphate pH 3 (75% *v/v*). Elution was done at a flow rate of 0.7 mL/min and the absorbance of the samples was read at 225 nm. The e.e was determined at final conversion and the calculation was done using the following equation:e.e (%) = ((*R*-MFG − *S*-MFG)/(*R*-MFG + *S*-MFG)) × 100,
where *R*-MFG and *S*-MFG are the concentrations (mM) of the *R*-isomer and *S*-isomer, respectively.

### 3.5. Stability of Novozym 435

Novozym 435 was incubated in non-reactive conditions in the presence of phosphate buffer and ChCl:urea 50% (*v/v*). At different times, samples were withdrawn and washed with water. The activity was measured as described previously and was expressed in terms of residual activity (ratio between final and initial activity).

## 4. Conclusions

The selectivity of Novozym 435 in the asymmetric hydrolysis of DMFG was highly improved through the use of high substrate concentrations and a DES-buffer mixture as reaction medium. Thus, the *R*-MFG was obtained with a high degree of purity (e.e = 99%) and moderate specific productivity. It should be noted that there is an important effect of the substrate concentration on the selectivity of Novozym 435, which is an asset that has not yet been adequately exploited in the asymmetric hydrolysis reaction.

## Figures and Tables

**Figure 1 molecules-24-00792-f001:**
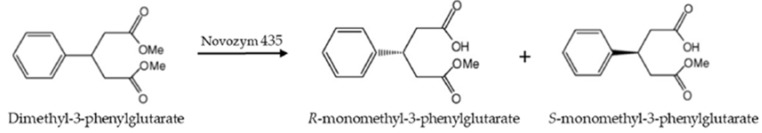
Asymmetric hydrolysis of dimethyl-3-phenylglutarate catalyzed by Novozym 435.

**Figure 2 molecules-24-00792-f002:**
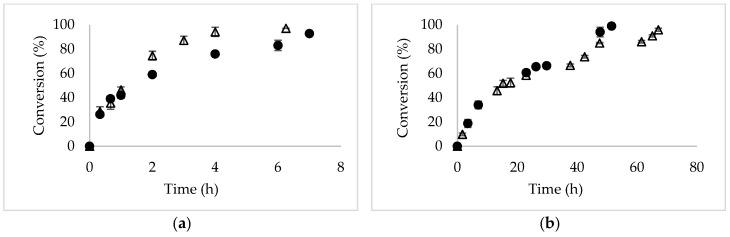
Reaction course of the asymmetric hydrolysis of DMFG (dimethyl-3-phenylgutarate) for the production of *R*-MFG (*R*-methyl-3-phenyl glutarate) catalyzed by Novozym 435. The reaction was performed at 170 rpm, pH 7 and 30 °C. (∆) phosphate buffer; (•) ChCl:urea 50% (*v/v*). (**a**) homogeneous system (0.45 mM of DMFG); (**b**) heterogeneous system (18 mM of DMFG).

**Figure 3 molecules-24-00792-f003:**
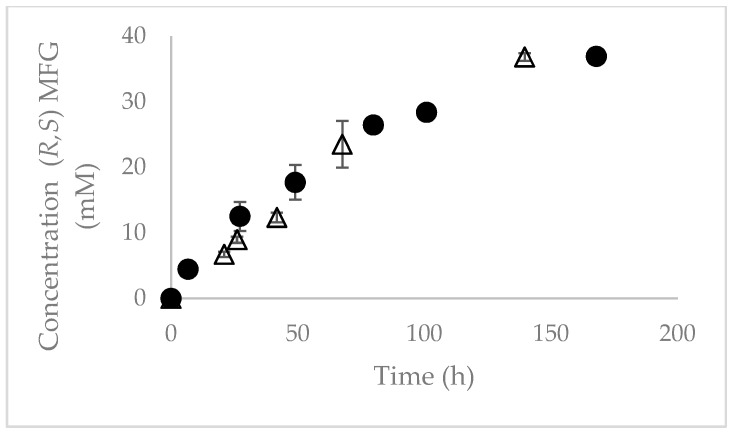
Production of *R*-MFG catalyzed by Novozym 435 in ChCl:urea 50% (*v/v*) reaction medium. The reaction was performed at 170 rpm, 30 °C and 36 mM (∆) and 54 mM (•) of DMFG.

**Figure 4 molecules-24-00792-f004:**
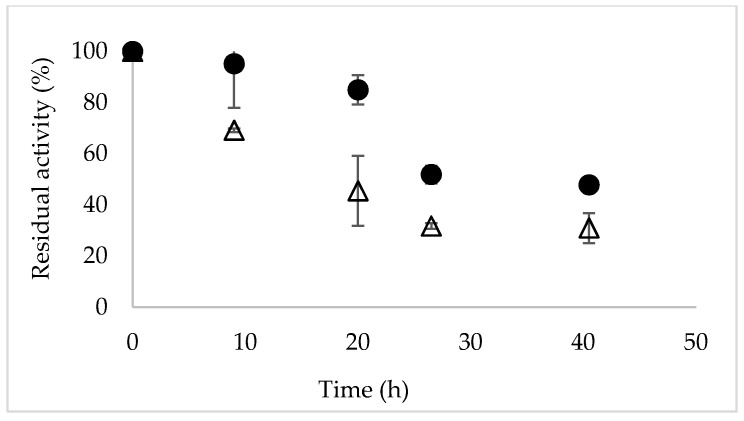
Courses of the inactivation of Novozym 435 at 170 rpm, pH 7 and 40 °C. (∆) phosphate buffer (•) ChCl:urea 50% (*v/v*).

**Table 1 molecules-24-00792-t001:** Specific initial reaction rate, enantiomeric excess (e.e) and specific productivity of the asymmetric hydrolysis of DMFG at pH 7 in different systems and reaction media.

Reaction Medium	DMFG (mM)	Reaction System	Specific Initial Reaction Rate (μmol Product/g Biocatalyst/min)	e.e^1^ (%)	Specific Productivity^1^ (g Product/g Biocatalyst/d)
Phosphate Buffer 100% (*v/v*)	0.45	homogeneous	2.2 ± 0.3	76	0.15 ± 0.01
	18	heterogeneous	6.0 ± 0.3	88	0.39 ± 0.02
ChCl:urea 50% (*v/v*)	0.45	homogeneous	2.1 ± 0.1	87	0.1 ± 0.01
	18	heterogeneous	6.2 ± 0.3	94	0.48 ± 0.01

^1^ Values were calculated at the maximum conversion.

**Table 2 molecules-24-00792-t002:** Specific initial reaction rate, enantiomeric excess and specific productivity in the asymmetric hydrolysis of DMFG in ChCl:urea 50% (*v/v*) at pH 7 and different substrate concentrations.

DMFG (mM)	Specific Initial Reaction Rate (μmoles Product/g Biocatalyst/min)	e.e^1^ (%)	Specific Productivity ^1^ (g Product/g Biocatalyst/d)
18	6.2 ± 0.3	94	0.48 ± 0.01
36	5.6 ± 0.3	98	0.46 ± 0.03
54	6.5 ± 0.3	99	0.68 ± 0.03

^1^ Values were calculated at the maximum conversion.

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
