# Peer review of "Increased Selectivity of Novozym 435 in the Asymmetric Hydrolysis of a Substrate with High Hydrophobicity Through the Use of Deep Eutectic Solvents and High Substrate Concentrations"

_molecules, 2019, doi:10.3390/molecules24040792_

Round 1
Reviewer 1 Report
This manuscript by Fredes et al. explores the effect of deep eutectic solvents (DES) on the catalytic activity of Novozym-435, asymmetric hydrolysis of dimethyl-4-phenylglutarate (DMFG). DES has been explored as a medium for enzymatic activation, and this study seems to adopts the method to demonstrate the promotion of Novozym-435 activity. The study itself is straightforward, and the results do seem to show the beneficial effect of DES on the enzymatic activity of DMFG hydrolysis. However, the results seem a bit incomplete, and needs further exploration before being considered for publication.
1. All the results do not show any statistical significance; no error bars and p values in Figures, standard deviation values in Tables. Without these information, it is hard to gauge the true effect of DES.
2. The plots in Figure 1 should be fitted with a mathematical model (e.g. Michaelis-Menten), and the conversion rates obtained from the fitting should be presented in Table 1. Table 1 does include the initial reaction rate, but does not show how those numbers were derived. This information should be obtained from the mathematical model.
Also, the plots in Figure 1 at first look do not seem to show significant difference between regular medium and DES medium seem. Including the fitted curves in the plots should help.
3. In Table 2, the amount of DMFG used for homogeneous condition was different for regular medium (0.45 mM) and DES medium (2.02). It is not appropriate to compare the two conditions with different monomer concentration. The author should explain why different DMFG concentrations were used. If possible, the experiment should be performed at the same concentration.
4. In Table 2, the authors explored the effect of DMFG concentration at heterogeneous condition. The authors should also explore the effect at homogeneous conditions as well.
Author Response
Point 1: All the results do not show any statistical significance; no error bars and p values in Figures, standard deviation values in Tables. Without these information, it is hard to gauge the true effect of DES.
Response 1: Reviewer is right. The error bars in figures 2, 3 and 4 were incorporated. Standar deviation in tables 1 and 2 also were incorporated.
Point 2: The plots in Figure 1 should be fitted with a mathematical model (e.g. Michaelis-Menten), and the conversion rates obtained from the fitting should be presented in Table 1. Table 1 does include the initial reaction rate, but does not show how those numbers were derived. This information should be obtained from the mathematical model.
Also, the plots in Figure 1 at first look do not seem to show significant difference between regular medium and DES medium seem. Including the fitted curves in the plots should help.
Response 2: Initial reaction rates were calculated from the initial slope of the product versus time curve.
We did not consider necessary in this case to apply the integral method to fit and calculate kinetic constants, since we are not sure, and it is not the objective of the work, whether the requirements are true: : absence of inactivation, absence of inhibition.
Point 3: In Table 2, the amount of DMFG used for homogeneous condition was different for regular medium (0.45 mM) and DES medium (2.02). It is not appropriate to compare the two conditions with different monomer concentration. The author should explain why different DMFG concentrations were used. If possible, the experiment should be performed at the same concentration.
Response 3: Reviewer is right. The experiment was carried out at different concentrations, because the objective was to compare both reaction media in saturated substrate conditions. For this reason, maximum substrate solubility was previously determined in the different reaction media (line 109). However, following the suggestion of the reviewer, the experiment at 0.45 mM of DMFG was performed and the results incorporated into the manuscript (Figure 2)
Point 4: In Table 2, the authors explored the effect of DMFG concentration at heterogeneous condition. The authors should also explore the effect at homogeneous conditions as well.
Response 4: The effect of substrate concentration in homogeneous conditions were not explored due to the low selectivity values that were obtained in that condition (Table 1).
Reviewer 2 Report
The authors described the effect of the eutectic mixture choline chloride/ urea on the selectivity and stability of Novozym 435. The results seem interestingly, but why the authors chose this particular DES among the many deep eutectic mixtures available? A comparison with the effect of other simple DESs I think it is important.
In the references, the articles of Capriati and coworkers about bio-catalysed reactions in DES based reaction media should be cited.
Finally, I think a reaction scheme would be useful to help the readers.
Author Response
Point 1: The results seem interestingly, but why the authors chose this particular DES among the many deep eutectic mixtures available? A comparison with the effect of other simple DESs I think it is important.
Response 1: Reviewer is right, many kinds of DESs have been developed with various compounds. To perform out this investigation, two simple DES based in choline chloride were considered. ChCl: Urea and ChCl: Glycerol were selected because they are among the most utilized DES with CALB lipase (immobilized and soluble) (Please see references 12, 15, 25). However, CHCl-glycerol was previously discarded because it showed low substrate solubility (0.87 mM of DMFG) in DES-Buffer mixture compared with ChCl: Urea (results not shown).
Point 2: In the references, the articles of Capriati and coworkers about bio-catalysed reactions in DES based reaction media should be cited.
Response 2: Several references were added with the aim improve the discussion and to clarify the novelty of work. Please see references 8, 12, 15, 25-26, 28-29. Results comparison were perform and improved, please see lines 50, 76, 78, 106, 116, 233.
Article: Brenna, D.; Massolo, E.; Puglisi, A.; Rossi, R.; Celentano, G.; Benaglia, M.; Capriati, V. Towards the development of continuous, organocatalytic, and stereoselective reactions in deep eutectic solvents. Beilstein J. Org. Chem. 2016, 12, 2620–2626 was incorporated in the manuscript.
Point 3: Finally, I think a reaction scheme would be useful to help the readers.
Response 3: Reviewer is right. The reaction scheme was introduced (Figure 1).
Reviewer 3 Report
The article describes the conversion of dimethyl-3-phenylgluterate by Novozym 435 in deep eutectic solvent (DES). The authors could show that the immobilised CalB is more stable in the in the investigated DES choline chloride / urea than in phosphate buffer and that up to 54 mM substrate concentration the enantiomeric excess increased by increasing substrate concentration.
One major drawback of this paper is that the author say in the material section, that all experiments were done in triplicates, but nowhere the in the figures are error barres displayed and in no table a significance of the results is shown. This fact makes the evaluation of the results difficult. Here also only one DES solvent was investigate for one substrate and no extension to different substrates or solvents was shown.
The second major drawback is that the results are not discussed properly with already published result. Only one citation can be found in the results and discussion section.
Especially there are in the review Xu et al. (Bioresour. Bioprocess. (2017) 4:34) around seven applications of Novozym 435 or CalB with the same DES system listed. Of course the applications differ, but some of these examples should be added to the introduction or discussion part.
Here also some specific comments:
Figure 1: that the time points for phosphate and DES system differ nearly always makes a comparison between the conversions difficult, error barres are missing
Table 1: for better comparison the reaction with 0.45 mM in ChCl/Urea 50 % should be add
Figure 2: Why is the product concentration dropping down after 160 h?
Page 8 line 294 Title of the Reference is missing
Page 9 line 302 Reference is not complete
Author Response
Point 1: One major drawback of this paper is that the author say in the material section, that all experiments were done in triplicates, but nowhere the in the figures are error barres displayed and in no table a significance of the results is shown. This fact makes the evaluation of the results difficult.
Response 1: Reviewer is right. The error bars in figures 1, 2 and 3 were incorporated. Standar deviation in tables 1 and 2 also were incorporated.
Point 2: Here also only one DES solvent was investigate for one substrate and no extension to different substrates or solvents was shown.
Response 2: Reviewer is right, many kinds of DESs have been developed with various compounds. To perform out this investigation, two simple DES based in choline chloride were considered. ChCl: Urea and ChCl: Glycerol were selected because they are among the most utilized DES with CALB lipase (immobilized and soluble) (Please see references 12, 15 and 25). However, CHCl-glycerol was previously discarded because it showed low substrate solubility (0.87 mM of DMFG) in DES-Buffer mixture compared with ChCl: Urea (results not shown).
Regarding the selected substrate, we decided to focus our attention on the DMFG, because it presents a very low solubility in aqueous medium and moderate values of enantiomeric excess, when Novozym 435 as biocatalyst is utilized. Besides, given the characteristics of the molecule we believe that the results obtained can be extrapolated to other hydrophobic substrates, in particular, to different diesters of phenyl glutaric acid.
Point 3:The second major drawback is that the results are not discussed properly with already published result. Only one citation can be found in the results and discussion section. Especially there are in the review Xu et al. (Bioresour. Bioprocess. (2017) 4:34) around seven applications of Novozym 435 or CalB with the same DES system listed. Of course the applications differ, but some of these examples should be added to the introduction or discussion part.
Response 3: Several references were added with the aim improve the discussion and to clarify the novelty of work. Please see references 8, 12, 15, 25-26, 28-29. Results comparison were perform and improved, please see lines 50, 76, 78, 106, 116, 233.
Article: Xu, P.; Zheng, G-W.; Zong, M-H.; Li, N.; Lou, W-Y. Recent progress on deep eutectic solvents in biocatalysis. Bioresour. Bioprocess 2017; 4:34. was incorporated in the manuscript.
Point 4: Figure 1: that the time points for phosphate and DES system differ nearly always makes a comparison between the conversions difficult, error barres are missing
Response 4: Reviewer is right. The error bars in figures 1, 2 and 3 were incorporated. Standar deviation in tables 1 and 2 also were incorporated.
Point 5: Table 1: for better comparison the reaction with 0.45 mM in ChCl/Urea 50 % should be add
Response 5: Reviewer is right. The experiment was carried out at different concentrations, because the objective was to compare both reaction media in saturated substrate conditions. For this reason, maximum substrate solubility was previously determined in the different reaction media (line 109). However, following the suggestion of the reviewer, the experiment at 0.45 mM was performed and the results incorporated into the manuscript.
Point 6:Figure 2: Why is the product concentration dropping down after 160 h?
Response 6: Reviewer is right, there is an error in figure 2 (now figure 3). We apologize for the error. The figure 3 was replaced and the concentration of 36 mM was incorporated to better validate the results obtained.
Point 7:Page 8 line 294 Title of the Reference is missing
Response 7: Reviewer is right. The reference was improved.
Chen, L.-Y.; Zaks, A.; Chackalamannil, S.; Dugar, S. Asymmetric synthesis of substituted 2-azaspiro-[3.5]-nonan-1-ones: An enantioselective synthesis of the cholesterol absorption inhibitor (+)-SCH 54016, J Org Chem 1996, 61, 8341-8343.
Point 8:Page 9 line 302 Reference is not complete
Response 8: Reviewer is right. The title was incorporated.
Maugeri, Z.; Domínguez de María, P. Benzaldehyde lyase (BAL)-catalyzed enantioselective CC bond formation in deep-eutectic-solvents-buffer mixtures. Journal Molecular Catalysis B: Enzymatic. 2014, 107, 120-12
Reviewer 4 Report
In this article it is described the influence of the reaction media (100 % phosphate buffer or choline chloride:urea/phosphate buffer 50%) and substrate concentration (from 0.45 to 54 mM) on the asymmetric hydrolysis of dimethyl-3-phenylglutarate catalyzed by Novozym 435, paying special attention to the enantiomeric excess and specific productivity of the reaction.
Optimization of the reaction conditions to achieve higher reaction conversion and enantiomeric excess is an important topic from the industrial point of view. Furthermore, exploring the applicability of new solvents in biocatalysis in very desirable to expand the variety of substrates that could be used and to modulate the enzyme properties.
However, I have some comments:
- Although I find the language understandable, the text of the manuscript should be check for typos and small grammar mistakes.
- The title “Increased selectivity of Novozym 435 in the asymmetric hydrolysis of substrates with high hydrophobicity through the use of deep eutectic solvents and high substrate concentrations” might be misleading since only one substrate is tested.
- A scheme in the introduction section showing the hydrolytic reaction would be helpful for the reader.
- There is no mention to the product of the hydrolysis of both methyl esters, were only mono-hydrolyzed products detected? Is there any reason for that?
- Line138 “The effect of substrate concentration on e.e has been seldom reported.” A reference there would be nice. Furthermore, this paragraph contradicts the following statement in the introduction “Despite these interesting results, increasing substrate concentrations may considerably reduce the enantiomeric excess, as shown by a decrease from 91% to 85% in the hydrolysis of diethyl-3-hydroxyglutarate catalyzed by Novozym 435 [9]. Therefore, the use of high substrates concentrations in the synthesis of chiral compounds has some drawbacks that need to be solved” (lines 51-55). Since in that case the same biocatalyst is used with a somehow similar substrate it would be nice to have a possible explanation of why the behavior shown is different.
- Regarding to M&M section, it is not clear how the reaction samples are taken and subsequently treated or how the substrate and product concentrations were calculated. Were the retention times compared to standards? Were they used any calibration curves or internal standards? Were the products identified by NMR?
- References 20 and 23 in the reference list lack some information.
- A reference and discussion of “Asymmetric hydrolysis of dimethyl 3-phenylglutarate catalyzed by Lecitase Ultra® Effect of the immobilization protocol on its catalytic properties” Enzyme and Microbial Technology 43 (2008) 531–536, should be included.
- Finally, my main concern is that in general I miss a deeper comparison of the results obtained in this study with those previously reported in the literature, highlighting the contribution of this investigation to the field.
Author Response
Point 1: Although I find the language understandable, the text of the manuscript should be check for typos and small grammar mistakes.
Response 1: Thanks for the suggestion, the manuscript was checked by a native English speaking colleague
Point 2: The title “Increased selectivity of Novozym 435 in the asymmetric hydrolysis of substrates with high hydrophobicity through the use of deep eutectic solvents and high substrate concentrations” might be misleading since only one substrate is tested.
Response 2: Thanks for the suggestion, the title of the article was modified.
Point 3: A scheme in the introduction section showing the hydrolytic reaction would be helpful for the reader.
Response 3: Reviewer is right. The reaction scheme was introduced (Figure 1).
Point 4: There is no mention to the product of the hydrolysis of both methyl esters, were only mono-hydrolyzed products detected? Is there any reason for that?
Response 4: In this work the presence of diacid was negligible (less than 2%). We believe that the hydrophobicity of the biocatalyst (Novozym 435) facilitates the partition of the monoster outside the active site of enzyme. This, based on the work of G. Fernandez-Lorente et al. (reference 31) where the hydrolysis of a dicarboxylic diester promote generate a free carboxylic acid which will be charged to pH 7. This charged compound will partition out of active site of the enzyme, which is a highly hydrophobic environment, formed by the hydrophobic areas of lipase and the hydrophobic support. This appears to virtually stop the reaction at the level of monoester
Point 5: Line138 “The effect of substrate concentration on e.e has been seldom reported.” A reference there would be nice. Furthermore, this paragraph contradicts the following statement in the introduction “Despite these interesting results, increasing substrate concentrations may considerably reduce the enantiomeric excess, as shown by a decrease from 91% to 85% in the hydrolysis of diethyl-3-hydroxyglutarate catalyzed by Novozym 435 [9]. Therefore, the use of high substrates concentrations in the synthesis of chiral compounds has some drawbacks that need to be solved” (lines 51-55).
Response 5: As the reviewer indicated, reference 10 was incorporated in the sentence. Besides, the paragraph “Despite these interesting results, increasing substrate concentrations may considerably reduce the enantiomeric excess, as shown by a decrease from 91% to 85% in the hydrolysis of diethyl-3-hydroxyglutarate catalyzed by Novozym 435 [9]. Therefore, the use of high substrates concentrations in the synthesis of chiral compounds has some drawbacks that need to be solved” (lines 51-55) was improved.
Point 6: Since in that case the same biocatalyst is used with a somehow similar substrate it would be nice to have a possible explanation of why the behavior shown is different.
Response 6: The reviewer's concern is very accurate. We believe that the differences are related to the structure of the substrate: DMFG has a phenyl group that confers a highly hydrophobic character to the molecule, therefore its interaction with the active site of the enzyme can be very different from that of diethyl-3-hydroxyglutarate. Proof of the above is that in the work of Jacobsen the isomer S is obtained, while in our work the isomer R.
Point 7: Regarding to M&M section, it is not clear how the reaction samples are taken and subsequently treated or how the substrate and product concentrations were calculated. Were the retention times compared to standards? Were they used any calibration curves or internal standards? Were the products identified by NMR?
Response 7: Reviewer is right. We regret the inaccuracies. The materials and methods section was improved.
Point 8: References 20 and 23 in the reference list lack some information.
Response 8: Reviewer is right. The reference was improved.
Chen, L.-Y.; Zaks, A.; Chackalamannil, S.; Dugar, S. Asymmetric synthesis of substituted 2-azaspiro-[3.5]-nonan-1-ones: An enantioselective synthesis of the cholesterol absorption inhibitor (+)-SCH 54016, J Org Chem 1996, 61, 8341-8343
Maugeri, Z.; Domínguez de María, P. Benzaldehyde lyase (BAL)-catalyzed enantioselective CC bond formation in deep-eutectic-solvents-buffer mixtures. Journal Molecular Catalysis B: Enzymatic. 2014, 107, 120-123.
Point 9: A reference and discussion of “Asymmetric hydrolysis of dimethyl 3-phenylglutarate catalyzed by Lecitase Ultra® Effect of the immobilization protocol on its catalytic properties” Enzyme and Microbial Technology 43 (2008) 531–536, should be included.
Response 9: According to the suggestion of the reviewer, the cited reference was included in the manuscript (reference 28).
Point 9: Finally, my main concern is that in general I miss a deeper comparison of the results obtained in this study with those previously reported in the literature, highlighting the contribution of this investigation to the field.
Response 9: Several references were added with the aim improve the discussion and to clarify the novelty of work. Please see references 8, 12, 15, 25-26, 28-29. Results comparison were perform and improved, please see lines 50, 76, 78, 106, 116, 233.
Round 2
Reviewer 1 Report
The revisions are adequate enough for publication.
Author Response
Thanks for your comments and suggestions, they were very helpful to improve our manuscript.
Reviewer 2 Report
The authors have replied to the most important issues, so I think the article is almost ready for the publication. Only I think that more interesting was the article of Capriati e Vitale (2017, 359, 1049-1057) where it is shown the effect on the stereoselectivity of a biocatalytic process in DES or water.
Author Response
Point 1: The authors have replied to the most important issues, so I think the article is almost ready for the publication. Only I think that more interesting was the article of Capriati e Vitale (2017, 359, 1049-1057) where it is shown the effect on the stereoselectivity of a biocatalytic process in DES or water.
Response 1: Reviewer is right, cited reference was incorporated (reference 28) with the aim of improving the discussion of manuscript.
Reviewer 3 Report
The authors changed all important points mentioned before.
But there are still some small mistakes in the text.
Line 39: ) must be deleted
Line 171: Figure 1 must be Figure 2
Line 216: there is something wrong with the sentence
Line 292: The x-axis of Figure 3 must be named in the right way: time (h) and is shown in a different style
Line 299: Figure 3 must be Figure 4
Author Response
Point 1: Line 39: ) must be deleted
Response 1: Reviewer is right, " ) " was removed.
Point 2: Line 171: Figure 1 must be Figure 2
Response 2: Reviewer is right, the number of the figure was modified
Point 3: Line 216: there is something wrong with the sentence
Response 3: Reviewer is right, the sentence was modificated
“However, in the present work an increase of substrate concentration exerts a positive effect on the selectivity of Novozym 435”
Point 4: Line 292: The x-axis of Figure 3 must be named in the right way: time (h) and is shown in a different style
Response 4: Reviewer is right, the x-axis of Figure 3 and the style of the text was modified
Point 5: Line 299: Figure 3 must be Figure 4
Response 5: Reviewer is right, the number of the figure was modified
Reviewer 4 Report
The authors addressed the issues mentioned before, however there are some points to be revised.
There are still typos, among them:
Some figures have a comma instead of a point as decimal separator.
Line 23 and throughout all the text: there should be a blank space between the figure and the Celsius degrees symbol. Moreover, this symbol should be consistent through the manuscript since now it is written in two different ways.
Line 37: “The currently, the…”
Line 39: a closing parenthesis without an opening one.
Line 42: “…strategies for the synthesis of pure enantiomers…”
Line 108: “The substrate dimethyl 3-phenylgutarate (DMFG), used…” the hyphen after “dimethyl” is missing.
Line 172: “…when the same DMFG of concentration was utilized.”
Line 216 “…may be related to the type of molecule and its…”
Materials and Methods: the first “p” in “pNP” is not always in italics.
Line 441: “Thus, the R-MFG…” the “R” should be in italics.
In figure 1, the enzyme’s name should be added on top of the reaction arrow.
Paragraph 232-238: I appreciate the explanation about the hydrolysis of only one out of the two ester groups of the molecule, nevertheless I find this paragraph difficult to read and understand.
Regarding to the discussion of the results, in addition to including the reference it would be helpful for the reader to have a sentence or two further explaining the cited results, and/or highlighting the difference with / novelty of those described in this work.
Author Response
Point 1: Some figures have a comma instead of a point as decimal separator.
Response 1: Reviewer is right, the numbers were modified
Point 2: Line 23 and throughout all the text: there should be a blank space between the figure and the Celsius degrees symbol. Moreover, this symbol should be consistent through the manuscript since now it is written in two different ways.
Response 2: Reviewer is right, a blank space between the figure and the Celsius degrees symbol was incorporated. Besides, Celsius degrees symbol was written in only ways.
Point 3: Line 37: “The currently, the…”
Response 3: Reviewer is right , word "the" was removed
Point 4: Line 39: a closing parenthesis without an opening one.
Response 4: Reviewer is right, " ) " was removed.
Point 5: Line 42: “…strategies for the synthesis of pure enantiomers…”
Response 5: Reviewer is right, sentence: “…strategies for the synthesis of pure enantiomers…” was modificated.
Point 6: Line 108: “The substrate dimethyl 3-phenylgutarate (DMFG), used…” the hyphen after “dimethyl” is missing.
Response 6: Reviewer is right, the hyphen was incorporated
Point 7: Line 172: “…when the same DMFG of concentration was utilized.”
Response 7: Reviewer is right, sentence was modificated.
Point 8: Line 216 “…may be related to the type of molecule and its…”
Response 8: Reviewer is right, sentence was modificated.
Point 9: Materials and Methods: the first “p” in “pNP” is not always in italics.
Response 9: Reviewer is right , the words were modificated
Point 10: Line 441: “Thus, the R-MFG…” the “R” should be in italics.
Response 10: Reviewer is right , the word was modificated
Point 11: In figure 1, the enzyme’s name should be added on top of the reaction arrow.
Response 11: The name of the enzyme in figure 1 was incorporated.
Point 12: Paragraph 232-238: I appreciate the explanation about the hydrolysis of only one out of the two ester groups of the molecule, nevertheless I find this paragraph difficult to read and understand.
Response 12: The indicated paragraph was improved to facilitate its reading.
Point 13: Regarding to the discussion of the results, in addition to including the reference it would be helpful for the reader to have a sentence or two further explaining the cited results, and/or highlighting the difference with / novelty of those described in this work.
Response 13: Reviewer is right, different sentences for support the results obtained were incorporated (lines 260-262 and 399-401).